# A Comparative Investigation on the Microstructure and Thermal Resistance of W-Film Sensor Using dc Magnetron Sputtering and High-Power Pulsed Magnetron Sputtering

**Jing Huan [1], Zhengtao Wu [1,2], Qimin Wang [1,*], Shihong Zhang [3] and Se-Hun Kwon [4]**

1    School of Electromechanical Engineering, Guangdong University of Technology, Guangzhou 510006, China; ztwu@gdut.edu.cn (Z.W.)
2    Guangdong Provincial Key Laboratory of Advanced Manufacturing Technology for Marine Energy Facilities, Guangdong University of Technology, Guangzhou 510006, China
3    Key Laboratory of Green Fabrication and Surface Technology of Advanced Metal Materials, Ministry of Education, Anhui University of Technology, Maanshan City 243000, China
4    School of Materials Science and Engineering, Pusan National University, Busan 609-735, Republic of Korea
*    Correspondence: qmwang@gdut.edu.cn; Tel.: +86-13802754560

**Abstract:** Traditional dc magnetron sputtering has a low ionization rate when preparing metallic thin films. With the development of thin film science and the market demand for thin film material applications, it is necessary to improve the density of magnetron-sputtered films. High-power pulsed magnetron sputtering (HiPIMS) technology is a physical vapor deposition technology with a high ionization rate and high energy. Therefore, in this work, HiPIMS was applied to prepare metallic tungsten films and compare the surface morphology and microstructure of metallic tungsten films deposited using HiPIMS and dc magnetron sputtering (dcMS) technology under different pulse lengths, as well as related thermal resistance performance, followed by annealing treatment for comparative analysis. We used AFM, SEM, XRD, and plasma characterization testing to comprehensively analyze the changes in the TCR value, stability, repeatability and other related performance of the metallic tungsten thin-film sensor deposited by the HiPIMS technology. It was determined that the thin film prepared by the HiPIMS method is denser, with fewer defects, and the film sensor was stable. The 400 °C annealed sample prepared using HiPIMS with a 100 μs pulse length reaches the largest recorded TCR values of $1.05 \times 10^{-3}$ K$^{-1}$. In addition, it shows better stability in repeated tests.

**Keywords:** HiPIMS; W film; thermal resistance; film sensor





## 1. Introduction

Among PVD-related technologies, magnetron sputtering technology is a relatively advanced thin film deposition technology. It has been widely used in production due to its advantages such as low deposition temperature and easy control of film thickness. However, ordinary magnetron sputtering technology has also shown its shortcomings in the industrial production process. The ionization rate in the production process is low, and the target material is etched unevenly, resulting in low utilization, usually only 20–30% [1]. In addition, the deposition process is unbalanced, producing plenty of growth defects in PVD thin films [2,3].

High-power pulsed magnetron sputtering (HiPIMS) can generate high-density plasma above the source cathode and increase the ionization rate of the sputtering source particles, which helps to control the kinetic energy and atomic migration of incident ions through the substrate bias [4–7]. HiPIMS technology has a very high metal ionization rate in the metal film deposition process, which benefits from the two characteristics of HiPIMS technology. The first is high-level pulse peak power, that is, high power, which is the pulse impact generated by high voltage [8,9]. The high current greatly improves the ionization rate of

the target, which generally exceeds the peak power of the traditional magnetron sputtering pulse by two orders of magnitude, but the emission of high-power pulses decreases the deposition rate. The second characteristic of HiPIMS technology is therefore the use of a low-level pulse duty cycle, which emits pulses with high frequency within the tolerable range of the electrode, and the duty cycle is generally 0.5% to 10%. Under high power, the proportion of metal ions increases, and some materials enter self-sputtering mode; that is, plasma is maintained by the ionization of sputtered neutral particles and secondary metal ions [10]. Compared with traditional dc magnetron-sputtering (dcMS) technology, the voltage of HiPIMS cathode can quickly rise above 1 kV in a pulse time and then rapidly decrease. Since the HiPIMS power supply can choose a smaller duty cycle, the temperature rise of the target is also lower than that of dcMS, which prevents the cathode from increasing its cooling requirements due to overheating.

Regarding related functional sensing films that are currently prepared on the market, thin-film temperature sensors prepared by the dcMS method have high resistivity, low sensitivity, and relatively poor stability [11,12]. In terms of resistivity, the film has a loose texture and many defects, which cannot meet the requirements for compactness. According to Matthiessen's rule [13,14], we know that plenty of defects and impurities in the film will enhance the scattering of electrons in the metal, leading to an increase in the resistivity of the films, which is one of the important factors that adversely affect the performance of the film sensor [15,16]. The sensitivity of the thin-film sensor is also one of the important indicators used to measure the performance of the sensor [11,17]. To improve the sensitivity of the thin-film sensor, increasing the temperature coefficients of resistance (TCR value) and the initial cumulative-resistance values ($R_0$) of the metallic films are required [18]. The increase in resistance $R_0$ of the film increases the consumption of metallic target, which not only increases the cost but also makes the film area too large or thin, which is not conducive to use. Therefore, the only way to improve the film thermal resistance sensor's sensitivity is to increase the temperature coefficient of resistance. Both reducing the resistivity and increasing compactness help increase the temperature coefficient of resistance of the metallic film [19–21]. Therefore, combining the above two points, we aim to explore the preparation of lower film resistivity and the defect treatment process after preparation, such as low-temperature annealing and other post-treatments.

A major advantage of HiPIMS technology is that it deposits denser and better-performing films through high ionization rates. The impact on the ionization rate is equivalent to the impact on the HiPIMS technology deposition process [4–7]. High-density plasma is the main factor for HiPIMS technology to maintain a high ionization rate during the deposition process, and the plasma density is related to the discharge parameters of the power supply, such as pulse length, frequency, peak current, and duty cycle during the deposition process. During film deposition using HiPIMS, the residence time of sputtered atoms is related to the pulse length and working pressure [22]. Adjusting the pulse length can change the residence time, thereby increasing the collision probability between sputtering atoms and increasing the ionization rate. The smaller the frequency, the more the capacitance can accumulate, which can increase the target's higher energy, thereby increasing the electron density, enhancing the ionization ability of particles, and increasing the ionization rate [23]. An increase in peak current increases the electron density and enhances the ionization ability of particles, thereby increasing the ionization rate [24–26]. In addition, the ion energy $E_i$ and ion-metal flux ratio $J_i/J_{Me}$ have significant effects on the microstructure and mechanical properties of the sputter-deposited thin films. Therefore, this work uses the pulse length as a variable and sets the difference in the HiPIMS group pulse–length–control experiment to explore the influence of changes in pulse length on deposited samples.

Since tungsten (W) has a large and stable temperature coefficient of resistance and high resistivity, the size of the W-based temperature sensor can be reduced by depositing metallic W film [27–29]. Tungsten has stable chemical and physical properties within the service temperature range, and the material has good reproducibility, a high melting



point, a stable structure, a resistance value that changes with temperature as a function of temperature, and good suitability for use as a thermal sensor material to measure temperature [30,31]. Metallic tungsten for dcMS and HIPIMS on the microstructure and thermal resistance of tungsten thin-film sensors are selected as the research material in this paper for a comparative study. We installed and operated bias-synchronized HiPIMS and dcMS in the sputtering equipment to grow pseudo-transition W films, which can detect the influence of metal ion concentration in different processes on film growth. A comparative investigation of the microstructure and thermal resistance of W-film sensor using dcMS and HiPIMS was conducted.

## 2. Materials and Methods

### 2.1. Film Deposition

The physical vapor deposition coating equipment used in this experiment has a vacuum chamber size of $760 \times 700$ mm$^2$. The equipment consists of a set of unbalanced magnetron sputtering sources, a set of high-power pulsed magnetron sputtering sources, and a set of matching magnetron targets. The rectangular ion source is combined with four sets of cathodic arc source ionization sources, which can install six different target materials at the same time and can realize the deposition of physical vapor deposition technologies. At the same time, the workpiece turret system of the coating machine AS500DMTXBH (Pro China Limited, Beijing, China) allows the turret to achieve revolution and independent rotation, which is convenient for directional coating. In the film deposition in this experiment, a dc magnetron power supply was used for the magnetron sputtering power supply, and a high-energy pulsed dc magnetron power supply was used for the high-power pulsed magnetron sputtering power supply.

In this experiment, a flat tungsten target of $449 \times 75$ mm$^2$ was used, mounted on a HiPIMS sputtering source, and a quartz block ($16 \times 16 \times 5$ mm$^3$) and a single crystal silicon wafer (100 crystal orientation) were used as the experimental substrate. The experimental substrate underwent a certain pretreatment to ensure the cleanliness of the surface of the substrate before the film was deposited and to improve the bonding force of the film substrate. The substrate in this experiment was ultrasonically cleaned with acetone and absolute ethanol in the pretreatment stage, dried with nitrogen, and put into the vacuum chamber; the various fixtures holding the substrate were also ultrasonically cleaned with acetone and absolute ethanol and blown dry. After the sample was cleaned, the number of experimental groups for depositing metal tungsten thin films using the HiPIMS technology was set to several groups.

After the vacuum chamber reached a base pressure of $5 \times 10^{-3}$ Pa, the temperature in the cavity was adjusted to 300 °C, and a glow cleaning was performed on the experimental substrate before vacuum coating to further remove particles on the surface of the substrate and residual contaminants. Glow cleaning was performed under an environment with a bias voltage of $-1000$ V and an Ar gas flow of 300 sccm, and the time was 30 min.

HiPIMS technology was used to deposit metal tungsten film. The amount of Ar gas was 70 sccm, and the pulse length was set to 50 μs, 100 μs, and 150 μs to perform three sets of HiPIMS technology deposition experiments. The discharge waveform of the HiPIMS was recorded with a Tektronix1000 oscilloscope (Tektronix 1000, Tektronix, Beaverton, OR, USA). The bias voltage was $-100$ V and the corresponding deposition times were 90, 60, and 45 min. For the most part, as the pulse length increases, the metal tungsten film deposition rate also increases in order to deposit metal tungsten films with a relatively close film thickness and a different deposition time.

After the deposition is completed, the cavity is cooled. After the temperature drops to room temperature, the cavity is opened, and the deposited substrate is taken out of the fixture and stored in a dust-free weighing paper package.

The experimental process of using dcMS technology to deposit metal tungsten film is as follows. The number of experimental groups is set to one group as a comparative experiment for HiPIMS technology deposition. In the dc magnetron sputtering process, we

used the same plane tungsten target as that of the HiPIMS deposition group, control the argon flow rate of 70 sccm, and deposit the metal tungsten film under the conditions of bias voltage –100 V, working pressure 0.5 Pa, cathode power 1 kW, and deposition temperature 300 °C. The deposition time is 60 min.

*2.2. Film Characterization*

In this paper, an ultra-high-resolution field emission scanning electron microscope (Hitachi SU8220, Hitachi, Tokyo, Japan) with a working voltage of 10.0 kV and a working distance of 10.0~20.0 mm was used to complete the characterization and analysis of the surface and cross-sectional morphology of the coating sample. The phase structure of the deposited and annealed coatings was characterized by X-ray diffraction (Bruker D8 advanced, Bruker, MA, USA). The scanning range was 20~90° in the normal mode, the step length was 0.02°, and the residence time per step was 1.0 s. The atomic force microscope used was the Tosca 400 atomic force microscope, produced by Anton Paar, Austria. The horizontal displacement sensitivity was 1 nm, the vertical displacement sensitivity was 0.8 nm, and the maximum scanning range was $100 \times 100$ μm$^2$. The contact between the needle and the sample was tapped. There are two kinds of type and contact type. In order to obtain more accurate roughness data, the scanning range was $100 \times 100$ μm$^2$ when calculating the roughness, and the average value was calculated through cycle tests. The size of the 3D topography map was $10 \times 10$ μm$^2$, and the grain distribution can be seen more clearly in this range. In this article, we used the RK2514A dc low-resistance tester(RK2514A, MeiRuiKe, Shenzhen, China) to measure resistance, and the measurement resistance range was 0.1~110 μΩ, whereas the test accuracy was 0.1%. In this paper, the four-wire method was used to measure the resistance. The metal copper film with relatively small resistance was prepared as a wire to connect with the metal W layer, and the external resistance meter was connected to the copper electrode to measure the resistance. The resistivity of the film can be calculated by combining the measured results of the resistance with the shape and size of the W film. When measuring the film resistance at high temperatures, in order to prevent the film from oxidizing in the air, the film and heating system were specially placed in the vacuum coating chamber. The vacuum coating chamber can provide a vacuum environment with a vacuum degree greater than $10^{-4}$ Pa and a temperature up to 500 °C, i.e., a high-temperature environment. The temperature in the vacuum chamber was detected in real time through an external thermocouple. The thermocouple's power cord and the wire for measuring the film resistance were connected to the outside through a sealed ceramic flange. All data related to thermal resistance were measured many times to attempt to eliminate the accidental factors caused by the experiment.

## 3. Results and Discussion

Figure 1 shows voltage and current of the W target with respect to HiPIMS pulse length. It was observed that as the pulse length increased from small to large, the duty cycle gradually increased. The target current and voltage changes in the HiPIMS process for preparing metal tungsten thin films with different pulse lengths is shown in the trend chart in Figure 1, where it can be observed that when a pulse length of 50 μs was applied, the peak current reached 463 A, and the bias voltage reached −845 V; when the pulse length was 100 μs, the peak current reached 234 A, and the bias voltage reached nearly −736 V; and when the pulse length was 150 μs, the peak current reached 198 A, and the bias voltage reached nearly −690 V. As the pulse length decreased, the peak current and peak power could be increased. With the increase in the matrix ion current, both the negative electron beam current and the positive ion beam current showed an increasing trend.

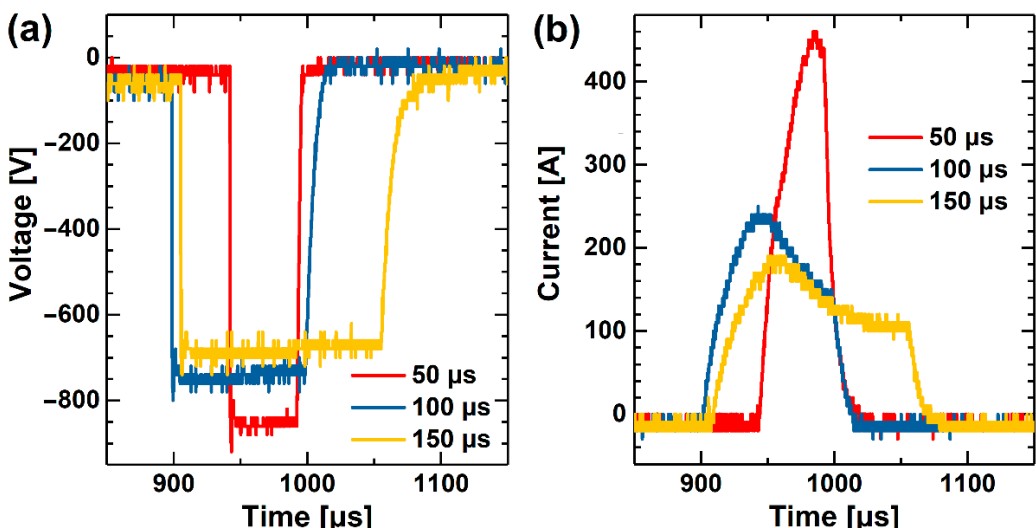

**Figure 1.** The (**a**) target voltage and (**b**) current of W with respect to HiPIMS pulse length.

The micromorphology of metal tungsten films prepared under different deposition processes is shown in Figure 2. The surface of the film is flat and has a strip-like structure, the stacking is regular and dense, the cross-sections all present a columnar crystal structure, the growth is continuous, and the film is dense. It can be observed that the metal W film prepared by the HiPIMS process had a denser cross-section than the film prepared by the dcMS process. The surface observation shows that the crystal grains are also smaller and the whole is denser. The results showed that the surface of the samples with pulse lengths of 50 and 100 μs was flatter and the cross-section was denser than that of the samples with pulse length of 150 μs. This may be because as the peak current increased, the energy during film deposition increased, the grain size of the film decreased, and the density of the W film increased [32,33]. This is because after the ionization rate was increased by decrease in pulse length, the film growth process and diffusion ability of the ions were strong, and the film was denser. The increase in ion energy also increases the nucleation rate, reduces the grain size, and increases the grain boundaries [34,35]. Therefore, the metal W film prepared by dcMS process shows a stronger particle sense, and the surface of the film appears uneven. The metal W thin films prepared by HiPIMS technology show flat and dense morphology.

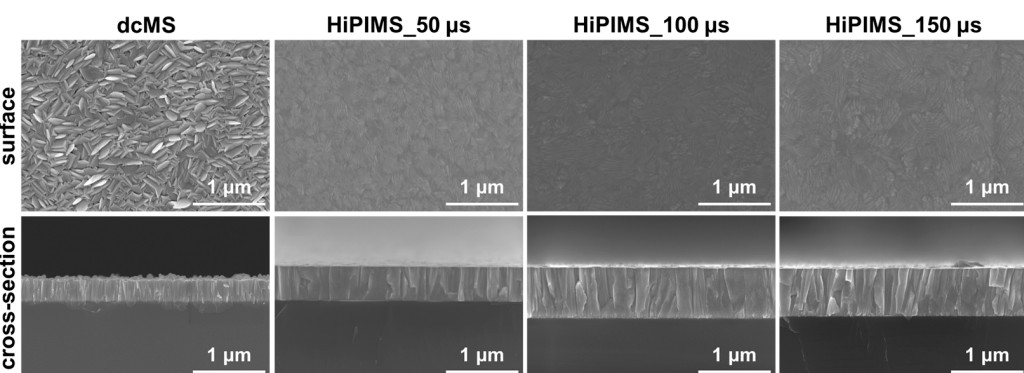

**Figure 2.** Surface and cross-section SEM images of W thin films with respect to HiPIMS pulse length.

Because the thickness of the film sample was small, this caused the resistance caused by the scattering of electrons on the surface of the film to account for a larger proportion of the total resistance, and the roughness of the film surface affected the resistivity of the film. To study the surface roughness of W thin films prepared under different deposition processes, this study used AFM to characterize the surface roughness of the samples.

Figure 3 shows the AFM morphology and roughness of W films prepared by different deposition processes. The scanning range was $10 \times 10 \ \mu m^2$. It can be seen that under the HiPIMS process conditions, the roughness of the prepared films varied with the pulse length. The extent of the increase gradually increased, but all the results are better than those of the roughness of the samples prepared using dcMS. Generally, the roughness is similar, indicating that the surface of several films has the same scattering ability for electrons, and the effect on the TCR value is basically similar [36–38].

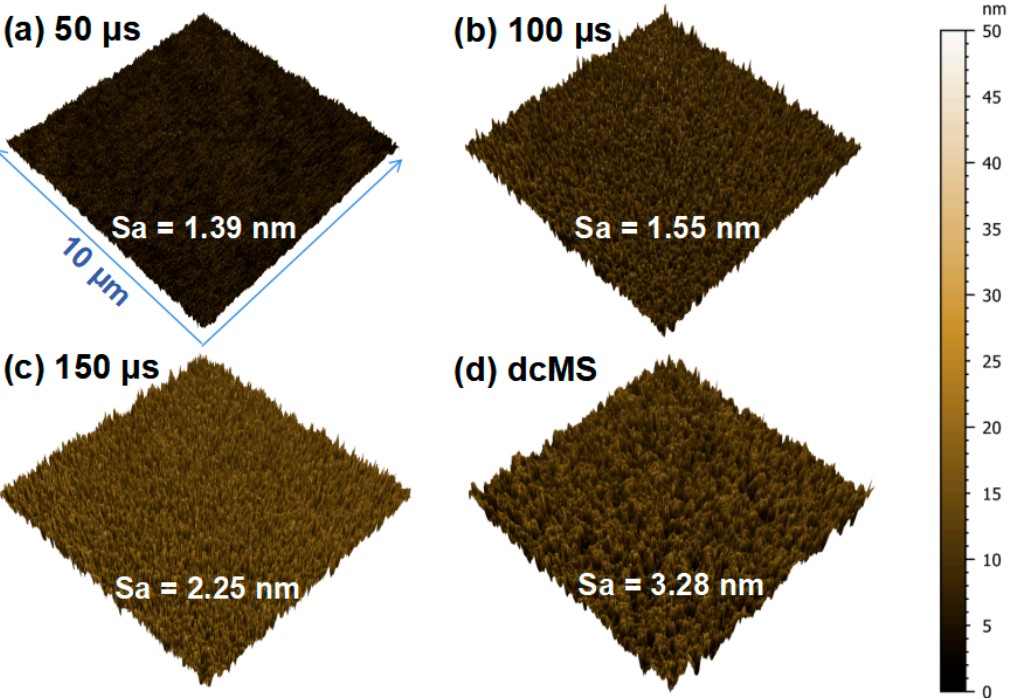

**Figure 3.** AFM morphologies and roughness of W thin films with respect to HiPIMS pulse length.

There are three crystal structures of W, and their structures and properties are different. Moreover, the magnetron-sputtered W film is in an unbalanced state, and several crystal structures may exist at the same time. This paper characterizes the phase structure of W films prepared using different deposition processes. Figure 4 shows the XRD patterns of metal tungsten films deposited using HiPIMS technology under different pulse length processes and metal tungsten films deposited using dcMS technology. The W films deposited under the process were all (110) oriented α-W, and no β-W appeared. The standard peak position of α-W tungsten is 40.32°. First, by comparing the deviation degree of the main peak of each metal tungsten with respect to the standard diffraction angle, the stress distribution of the metal tungsten film material can be obtained. The peaks of metal tungsten films deposited using HiPIMS technology with lengths of 50 μs and 100 μs are on the left side of the line of diffraction angle, indicating that the stress was compressive stress, and the metal tungsten film was deposited using dcMS technology and HiPIMS technology with a pulse length of 150 μs. The peak is near the line of the diffraction angle, and the deviation is small; that is, the internal stress is small. The peak height of the samples prepared by the HiPIMS process in the XRD graph is higher, indicating that the orientation degree is better than that of the dcMS process. In addition, it was determined that the full widths at half maximum of tungsten film prepared by HiPIMS are larger than dcMS, indicating that the HiPIMS-W films have refined grain sizes. It shows agreement with the SEM observation. The pictures therefore confirm each other.

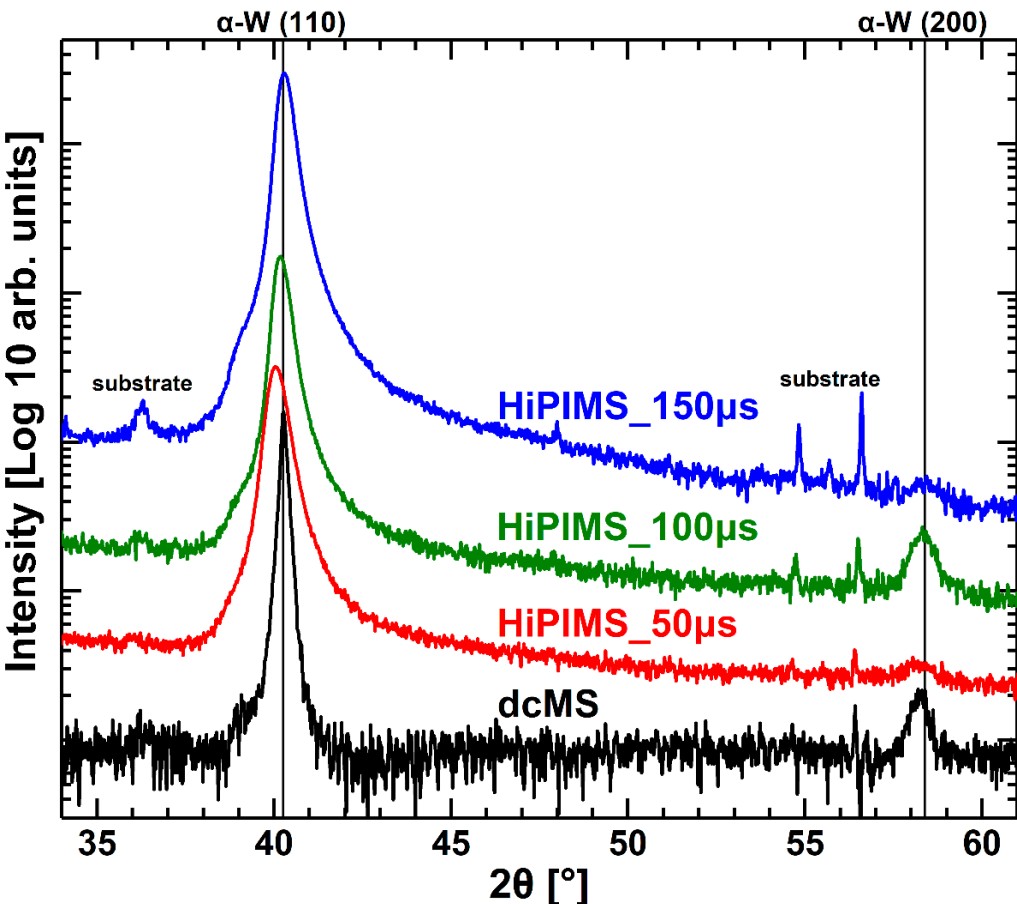

**Figure 4.** XRD spectra of W thin films prepared by dcMS and HiPIMS with varying pulse widths.

Figure 5 shows the thermal resistance curve of the W film. The resistance of the film increases with the increase in temperature, indicating that the prepared W film sensor has a positive temperature coefficient. This index is directly related to the sensitivity of the film [39]. The thermal resistance TCR of films prepared by different processes is not the same. In terms of TCR value, the sample prepared using HiPIMS with a 100 μs pulse length was the highest, reaching $1.05 \times 10^{-3}$ K$^{-1}$, followed by a 150 μs pulse length, the TCR value of the sample prepared with a pulse length of 50 μs was higher, and the sample prepared with dcMS had the lowest TCR value at only $5.43 \times 10^{-4}$ K$^{-1}$, and the performance was the worst. This may be because for metal W, the pulse length of 50 μs is compared with that of 100 μs. The pulse length of the HiPIMS process brought higher energy when the film was formed, which led to an increase in the density of the W film and a reduction in the scattering of electrons by the grain boundary. When the density of the film increased, the scattering of electrons by the grain boundary decreased, the restraining effect of the grain boundary on the TCR value decreased, and the TCR increased [40,41]. As far as the stability of the thin-film sensor is concerned, we determined that from the cycle measurement results, the thermal resistance data prepared by the dcMS process and the 150 μs pulse length HiPIMS process had the worst linearity, and the stability performance of the TCR was also the worst. In contrast, the thermal resistance curves of the W film prepared by the HiPIMS process with a pulse length of 50 μs and a pulse length of 100 μs have a linearity greater than 90% after linear fitting. The linearity is good, and the TCR value of cycle measurements is relatively stable.

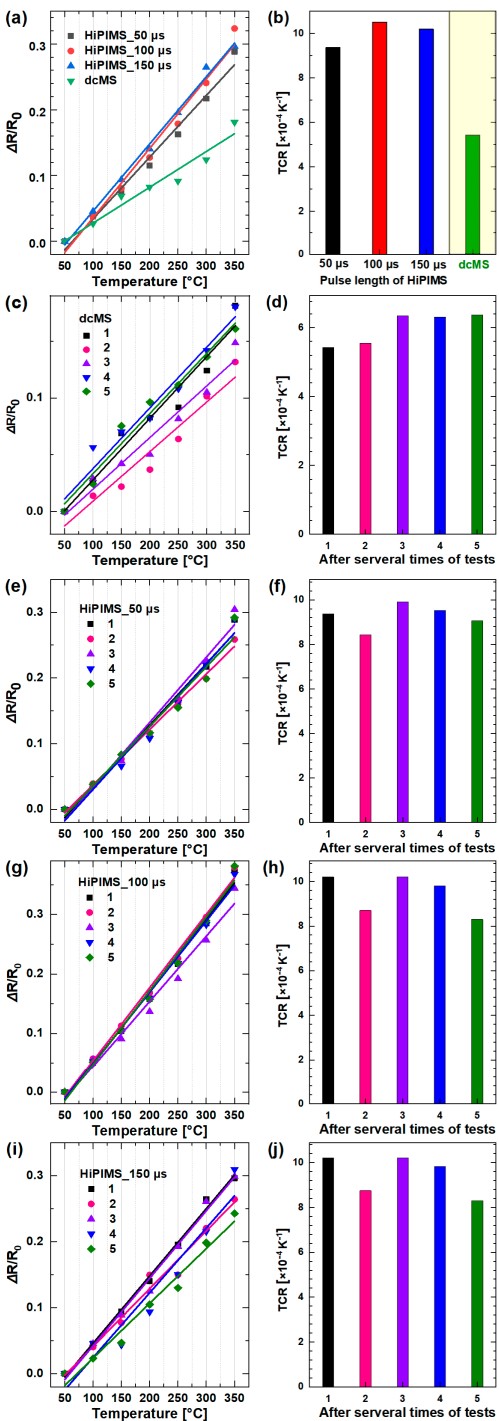

**Figure 5.** Thermal resistance test results of W thin films prepared by dcMS and HiPIMS. (**a**) Variation of electrical resistivity $\Delta R/R_0$ of W thin films with respect to temperature, (**b**) TCR values of W thin films. (**c**) Variation of electrical resistivity $\Delta R/R_0$ of dcMS-W thin film with respect to test numbers, (**d**) TCR values of dcMS-W thin film with respect to test numbers. (**e**) Variation of electrical resistivity $\Delta R/R_0$ of 50 μs-pulse-length HiPIMS prepared W thin film with respect to test numbers, (**f**) TCR values of 50 μs-pulse-length HiPIMS prepared W thin film with respect to test numbers. (**g**) Variation of electrical resistivity $\Delta R/R_0$ of 100 μs-pulse-length HiPIMS prepared W thin film with respect to test numbers, (**h**) TCR values of 100 μs-pulse-length HiPIMS prepared W thin film with respect to test numbers. (**i**) Variation of electrical resistivity $\Delta R/R_0$ of 150 μs-pulse-length HiPIMS prepared W thin film with respect to test numbers, (**j**) TCR values of 150 μs-pulse-length HiPIMS prepared W thin film with respect to test numbers.

In addition to the large resistance temperature coefficient and linearity, the performance requirements of thermal resistance materials also include the need to have a stable structure to ensure that no changes in composition and structure occur during high- and low-temperature cycles and that they have good cycle stability and repeatability. In order to test the repeatability of the thermal resistance coefficient of the W film prepared after the optimization of the process, the W film was heated to 350 °C in a vacuum environment, and after cooling, the stability was repeated several times. Figure 6 shows the cross-sectional SEM morphology of the W film after repeated tests. The surface of the deposited W film was flat, and there were no prominent crystal grains, but after heating cycles, there were protruding W crystal grains on the surface of the W film. It appears that as the surface became uneven, the size of these protruding grains became significantly smaller than the grain size of the deposited W film, indicating that the W film recrystallized slightly at high temperatures. Due to the short heating time, only a small amount of the W film was produced on the surface. After the high- and low-temperature cycle was repeated five times, the number of recrystallized grains on the surface of the film increased, and the surface of the W film became rougher. In addition, the thickness of the W film hardly changed after the thermal resistance test with high- and low-temperature cycles.

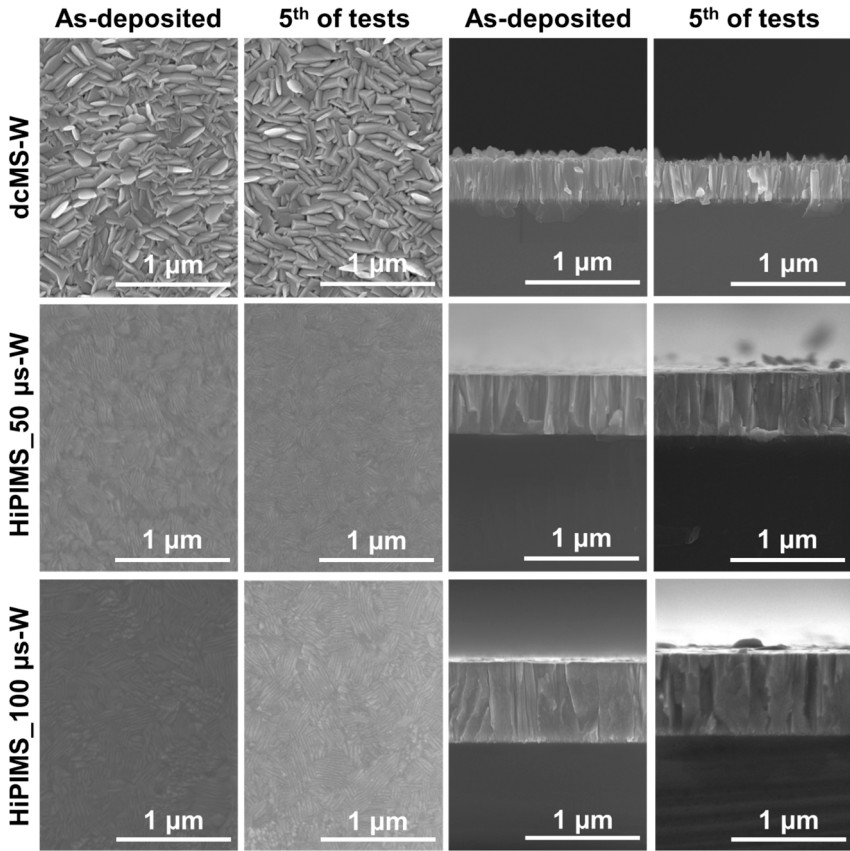

**Figure 6.** Surface and cross-section SEM images of W thin films prepared by dcMS and HiPIMS after cycle thermal resistance measurements.

According to observations, the cross-section of the film prepared by the HiPIMS process was still denser than that of the film prepared by the dcMS process, and the surface observation shows that the crystal grains were also smaller and the whole was denser.

The surface morphology and roughness of the W film after cycle measurements are shown in Figure 7. The surface roughness of the W film was slightly increased after cycle measurements because after the grains recrystallized, the surface began to become uneven.

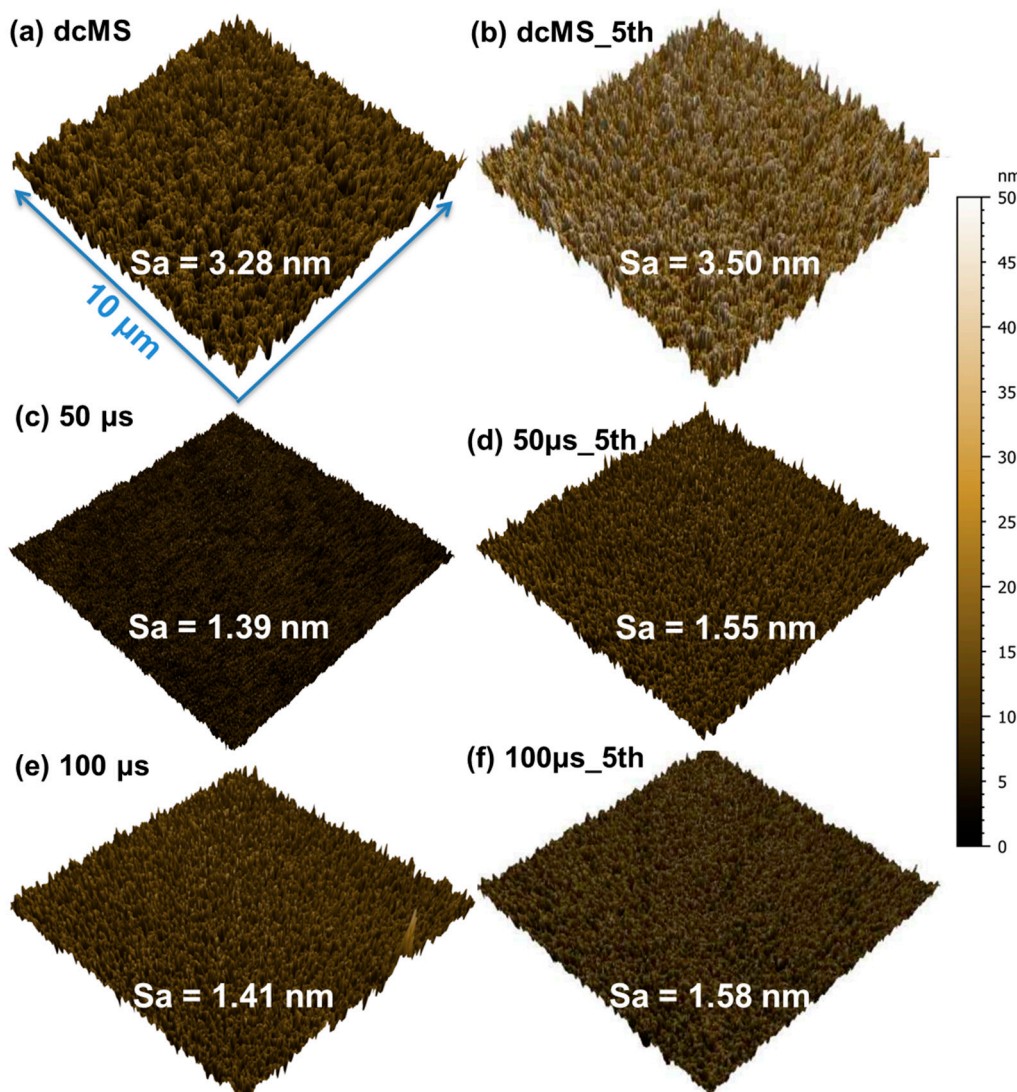

**Figure 7.** AFM morphologies and roughness of W films prepared by dcMS and HiPIMS after cycle thermal resistance measurements.

The XRD pattern of the W film only has the diffraction peak of body-centered cubic $\alpha$-W, as shown in Figure 8, which shows that the film did not undergo a phase change after cycle thermal resistance tests.

During the thermal resistance test, we determined that the process of increasing temperature causes low-temperature annealing in the heating of the film. The inside of the W film was recovered, with a small amount of recrystallization, and the crystal defects were reduced, except for the three sets of experiments prepared by the HiPIMS process with 150 μs pulse length. The TCR gradually began to become stable. In order to further study the influence of high temperature on the microstructure and thermal resistance properties of W thin films, and to obtain W thermal resistance thin films with good repeatability, the W thin films were annealed in this paper. The sample prepared by HiPIMS process with 150 μs pulse width has poor linearity of data fitting and poor stability of TCR test data in the thermal resistance test. Therefore, in the following annealing experiment, the film sensors prepared by dcMS, HiPIMS with pulse widths of 50, and 100 μs were selected for subsequent annealing treatments.

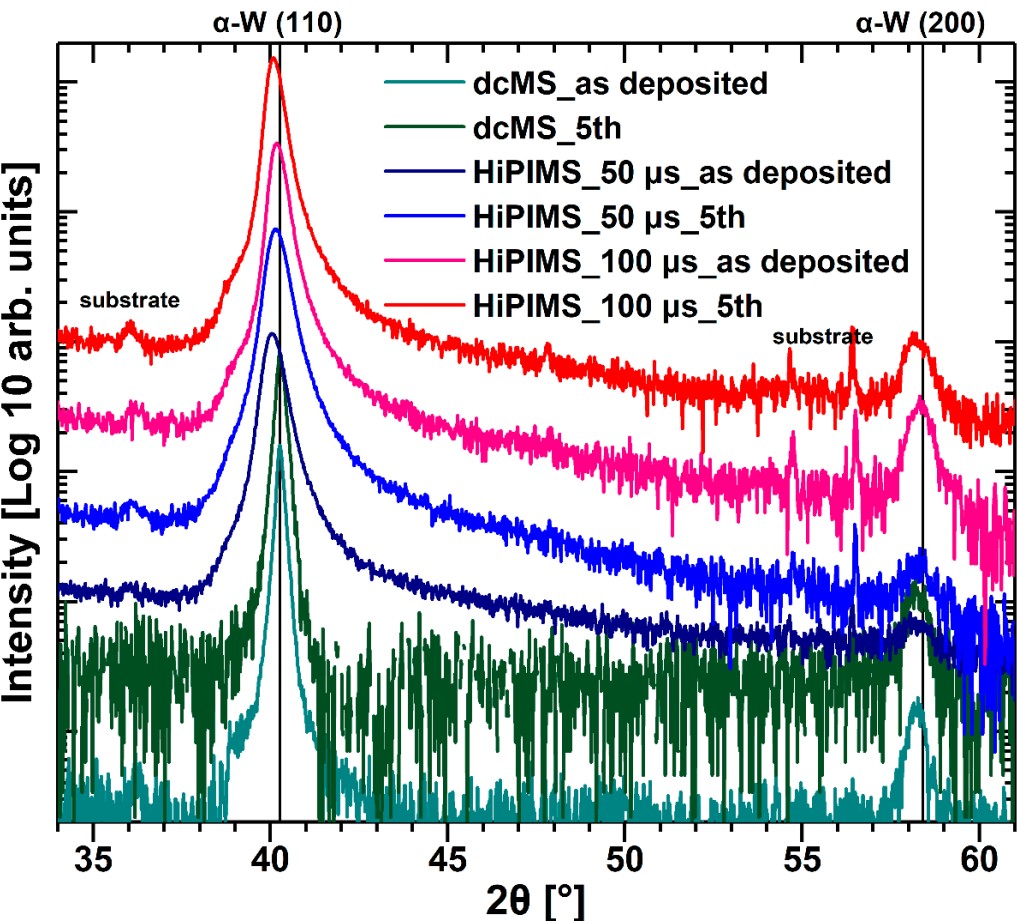

**Figure 8.** XRD spectra of W thin films prepared by dcMS and HiPIMS with 50/150 μs pulse widths after five times of thermal resistance tests.

Figure 9 is a cross-sectional SEM image of the W film after vacuum annealing at 500 °C for 0~1 h. The surface of the deposited W film was flat, and fine recrystallized grains appeared on the surface after annealing. The recrystallized grains on the surface grew after annealing for 60 min. The melting point of W metal is very high, and its recrystallization temperature is more than 1000 °C [42,43]. The grain size of the W film was much smaller than that of the bulk W and was nanocrystalline. Furthermore, the film deposited by magnetron sputtering technology had a large internal stress, and a small number of sputtering gas Ar atoms were incorporated [44,45]. These factors caused the W film to be far from the equilibrium state, and the recrystallization temperature was lowered.

The surface morphology and roughness of the W film after annealing are shown in Figure 10. After annealing, recrystallized grains began to appear on the surface of the W film, and the surface became uneven. As the annealing time increased, the surface protrusions increased and the recrystallized grains increased. After annealing for 60 min, the roughness remained basically unchanged. The XRD pattern of the W film only has the diffraction peaks of body-centered cubic α-W, as shown in Figure 11, where there is no phase change during the annealing process.

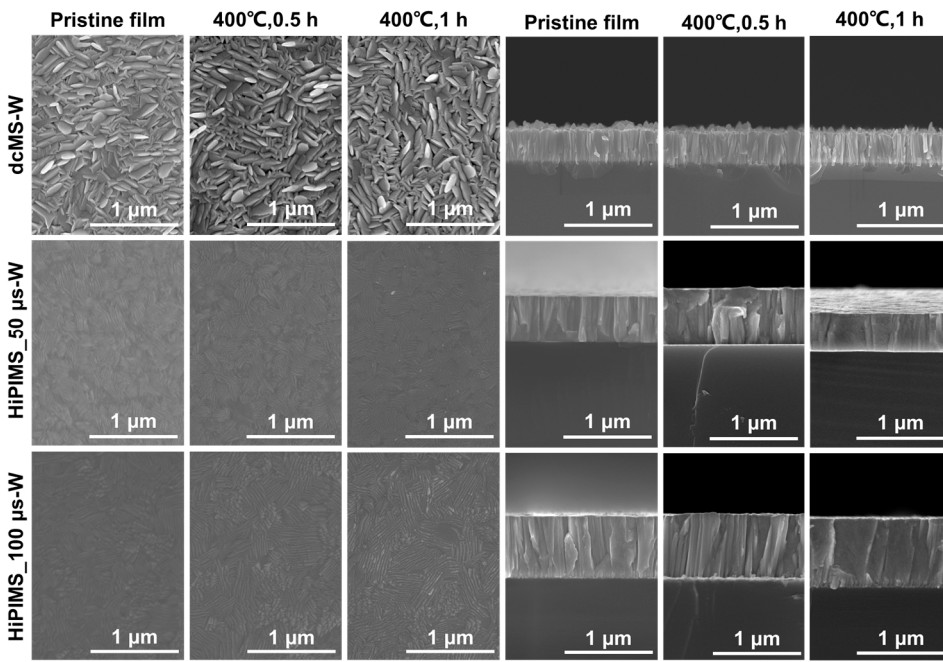

**Figure 9.** Surface and cross-sectional SEM images of W thin films after vacuum annealing at 400 °C for 0.5 and 1 h.

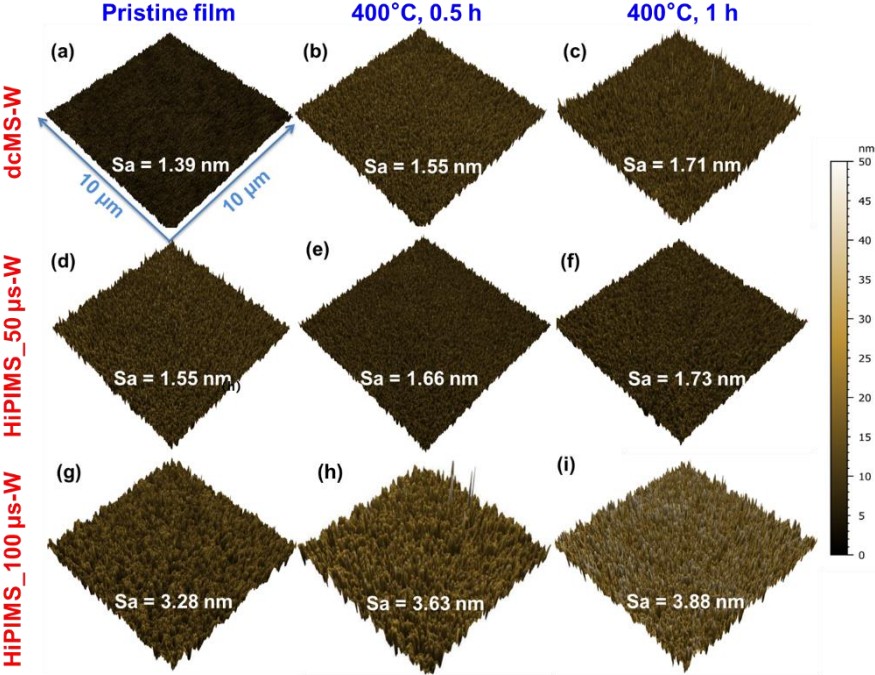

**Figure 10.** AFM morphologies and roughness (Sa) of W thin film surface after vacuum annealing at 400 °C for 0.5 and 1 h. (**a**) Pristine, (**b**) 400 °C/0.5 h, and (**c**) 400 °C/1 h annealed W films deposited by dcMS. (**d**) Pristine, (**e**) 400 °C/0.5 h, and (**f**) 400 °C/1 h annealed W films deposited by HiPIMS with pulse length of 50 μs. (**g**) Pristine, (**h**) 400 °C/0.5 h, and (**i**) 400 °C/1 h annealed W films deposited by HiPIMS with pulse length of 100 μs.

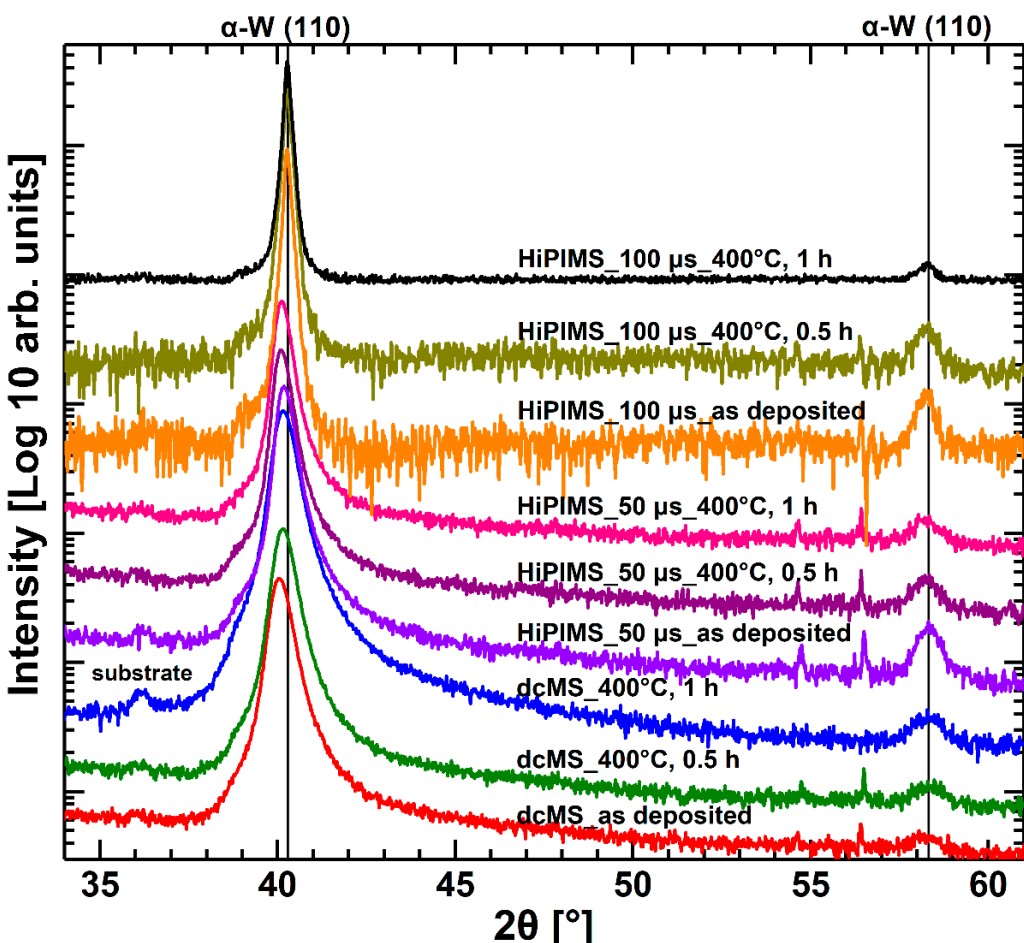

**Figure 11.** XRD spectra of W thin films after vacuum annealing at 400 °C for 0.5 and 1 h.

After annealing, the inside of the W film recovered and recrystallized. For a long time, we tested the thermal resistance film that had been vacuum annealed for 1 h. Figure 12 presents a linearity characterization graph showing the thermal resistance curve of the W film before and after annealing, as well as the TCR and linear fitting. The TCR of the W film prepared by the dcMS method increased from $5.43 \times 10^{-4}$ to $12.8 \times 10^{-4}$ K$^{-1}$ after annealing. After annealing, the W film recrystallized and crystal defects were greatly reduced, while the TCR increased. We continued to repeat the thermal resistance test for the annealed W film several times to check its structural stability and repeatability of thermal resistance. The thermal resistance curves of the annealed W film basically overlapped for five consecutive tests, and the TCR was basically the same, including after fitting. The linearity also improved, indicating that the annealed W film has good thermal resistance and the structure tends to be stable. The improvement in the TCR value of W films prepared under the conditions of 50 μs pulse length and 100 μs pulse length HiPIMS process, i.e., $11.5 \times 10^{-4}$ and $8.42 \times 10^{-4}$ K$^{-1}$, respectively, is not as high as that of samples prepared using dcMS method, and even the thin films prepared under 100 μs pulse length had some TCR after annealing treatment. However, in terms of data repeatability and the linearity of thermal resistance changes, these values are still far superior compared to those of the films prepared using the dcMS process, among which there is good stability. In other words, the thin-film sensor prepared under the condition of a 100 μs pulse length performed best.

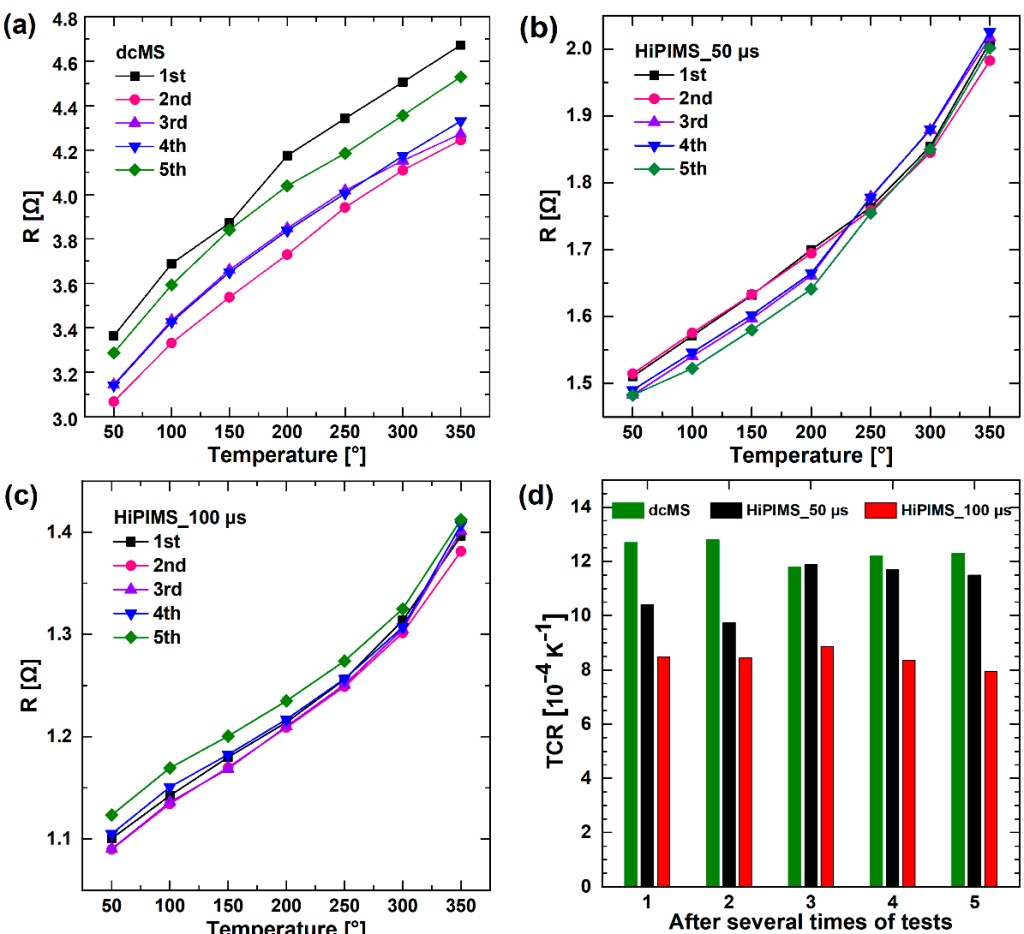

**Figure 12.** Thermal resistance test results of W thin films after vacuum annealing at 400 °C for 1 h. (**a**) Electrical resistivity of (**a**) dcMS, (**b**) 50 μs-pulse-length-HiPIMS, and (**c**) 100 μs-pulse-length-HiPIMS prepared W thin films with respect to temperature and test numbers, (**b**) electrical resistivity of dcMS-W thin film with respect to temperature and test numbers. (**d**) TCR values of W thin films prepared by dcMS and HiPIMS after vacuum annealing at 400 °C for 1 h.

The experiments using two different processes, i.e., HiPIMS and dcMS, to prepare metal W films show that by improving the process, we can prepare films with a smoother surface, higher density, and better uniformity than dcMS through the HiPIMS process. In terms of microscopic morphology, HiPIMS coating has a smooth surface with dense columnar crystals in the cross-section, and the crystal grains grew perpendicular to the substrate; the dcMS coating surface was rough, with holes between columnar crystals; and the crystal grains grew at a certain angle along the surface of the substrate.

Because the plasma density in front of the traditional dc magnetron target was low, and because the average electron energy was lower than the ionization potential of the sputtered atoms, only a small part of the sputtered atoms became ions through collision [46]. The energy of plasma and electrons of high-power pulsed magnetron sputtering was equivalent to that of a dc magnetron, but the ion flux was 2–3 orders higher. This means that when sputtered atoms pass through the critical electron collision ionization zone with cycle collisions, the probability of ionization greatly increases [46,47]. In the thin film deposition process within the HiPIMS process, a large amount of ion current accelerates to the substrate under the action of the substrate bias. On the one hand, the bombardment can remove the loose surface structure, and on the other hand, it can interrupt the penetrating columnar crystals and promote crystallization. Particle refinement and recrystallization improve the uniformity and compactness of the coating. Studies by Alami et al. [48] reported that when the average current is kept the same, the ion bombardment increases with the increase

in the peak current of the target, and the density and surface roughness of the coating also changes, increasing the target. With peak current, a porous columnar structure (and magnetron sputtering), dense columnar crystal, and amorphous structure are obtained in sequence.

In this study, through XRD detection and analysis, it was confirmed that after cycle tests and after heat treatment, the prepared W films did not undergo a phase change, and the films prepared using dcMS and HiPIMS methods showed a certain degree of stability in phase structure. Before and after annealing, the film prepared using the dcMS process experienced an increase in surface roughness from 3.28 nm to 3.88 nm, while the film prepared using the HiPIMS process, for example, the film prepared under the condition of 100 μs pulse length, experienced surface roughness. The roughness increased from 1.55 nm to 1.73 nm. This is because with repeated testing and annealing treatment, the surface of the film continuously precipitated crystal grains and recovery and recrystallization occurred, so the surface roughness of the film gradually increased and it became larger.

As far as the thermal resistance performance of the thin-film sensor is concerned, it is obvious that the W thin film prepared by the HiPIMS method exhibits better comprehensive performance. In this context, we need to discuss the generation and influencing factors of resistance. The scattering mechanism of electrons in metal has two main factors. One is the scattering caused by the thermal vibration of the lattice, which is called lattice scattering [49]. The other is the scattering caused by lattice defects or impurities [50]. Therefore, in the context of energy band theory, "resistance" refers to the transition of an electronic state caused by the destruction (or deviation) of the strict periodic field of the crystal. Different kinds of scattering have different contributions to resistance. When there are impurities and atomic thermal vibration at the same time, the resistance of the metal is the sum of these two parts of resistance. Under an extremely low temperature condition, the resistance caused by dislocations and phonons is extremely low. Only the resistance caused by impurities remains at this time, which is not connected with temperature but depends on the number of impurity atoms and impurity types contained in the metal. This resistance is called the "residual" resistance [51]. It can be seen that the relationship between the resistivity $\rho$ of metal and the temperature T is basically determined by the resistance contributed by lattice scattering.

At a normal temperature, $T > \theta_p$ ($\theta_p$ is the Debye temperature of the metal, generally 200~400 K) and the numbers of phonons corresponding to the grid wave increase as the temperature T increases. The greater the number of phonons, the more the electrons are scattered, so the metal resistivity $\rho$ is proportional to the temperature T [52,53]. At a low temperature, $T < \theta_p$, the probability of electron scattering (or the number of collisions per unit time) is determined by two factors: the phonon concentration and the effective number of collisions (that is, the number of scatterings required for each collision).

Due to the low temperature and particle energy of traditional dcMS, the film-base bonding force is insufficient, and the coating occurs mostly in a low-temperature phase and non-equilibrium phase, and there are many defects and impurities in the film, which limits its industrial application. According to Anders's structural area model [3], the energy shortage caused by low temperature can be compensated by high-energy ions. The HiPIMS technology has high ion energy and is simple and controllable, which compensates for the problem of low energy in the dcMS process [45,54]. The sample preparation is dense and there are few defects, thus achieving better results.

As far as the samples from the HiPIMS group are concerned, the overall performance of the W film prepared under the condition of 100 μs pulse length was the best. Combined with the experimental data of the oscilloscope, it is considered from the energy point of view because of the preparation of the metal W film and the ion energy. Additionally, the ion to metal flow ratio $J_i/J_{Me}$ has an important influence on the microstructure and mechanical properties of sputtered W films [55–57]. The energy under the condition of 100 μs pulse length is the most suitable preparation parameter among the three sets of pulse length at 150 μs. For the deposition of metal tungsten film in HiPIMS technology under the

condition of a long pulse length, the pulse length parameter setting was excessively high, resulting in lower peak current and lower ion energy during the deposition process, so the deposited film had more surface defects and higher density. The energy deposited under the condition of 50 μs pulse length was excessively high, which also adversely affected the performance of the film.

## 4. Conclusions

In summary, we used two different PVD processes, HiPIMS and dcMS, to prepare metallic W thin films as the sensitive layer of the temperature sensor, and we attempted to change the pulse length parameters of the HiPIMS power supply to set up different groups of control experiments. In the follow-up, we also attempted annealing treatment for the analysis to conduct a comparative study on the microstructure and thermal resistance of the tungsten thin-film sensor. Using W as the thermal resistance material, a thin-film temperature sensor was constructed, and the deposition process and parameters on the microstructure control law of the W thin film were studied. The microstructure of the W thin film was changed, and the influencing law of the thermal resistance effect of the sensing thin film was explored.

The results indicate that by comparing the W film deposited using HiPIMS with the film deposited using dcMS, in terms of surface morphology and structure, HiPIMS technology had smaller crystal grains than the film deposited using dcMS. HiPIMS-W films have lower roughness, fewer surface defects, and higher density compared to dcMS ones. In terms of thermal resistance performance, the HiPIMS-W films have better thermal stability, and the linearity and stability of resistance with temperature changes were better than those when using dcMS technology. The metal tungsten films deposited using HiPIMS with varying pulse lengths were compared in terms of surface morphology and structure; we determined that as the pulse length increased, the grain size gradually increased and the amounts of surface defects increased. In terms of thermal resistance performance, combining the initial TCR and the stability of cycle measurements, the thin-film sensor prepared under the condition of 100 μs pulse length has the best performance, with TCR value reaching $1.05 \times 10^{-3}$ K$^{-1}$, and the repeatability of cycle tests was excellent. After annealing treatment, we determined that the heat treatment process greatly improves the thermal resistance of the film prepared by the dcMS method. The TCR value increases from $5.43 \times 10^{-4}$ to $12.8 \times 10^{-4}$ K$^{-1}$, and the thermal resistance is stable. The HiPIMS films keep their excellent initial deposition characteristics. The TCR values change slightly as the annealing temperature or time increases, showing good stability and test repeatability. As far as the HiPIMS film temperature sensor is concerned, the annealing treatment can be completely omitted, thereby greatly saving its production time and material cost.

**Author Contributions:** Conceptualization, Z.W. and Q.W.; methodology, J.H.; software, J.H.; validation, Z.W., J.H. and Q.W.; formal analysis, J.H.; investigation, Z.W.; resources, J.H.; data curation, Q.W.; writing—original draft preparation, Z.W.; writing—review and editing, S.Z. and S.-H.K.; visualization, Z.W.; supervision, Q.W.; project administration, Q.W., S.Z. and S.-H.K.; funding acquisition, Q.W. All authors have read and agreed to the published version of the manuscript.

**Funding:** This research was funded by the National Key Research and Development Project of China (2017YFE0125400) and the National Natural Science Foundation of China (51901048).

**Institutional Review Board Statement:** Not applicable.

**Informed Consent Statement:** Not applicable.

**Data Availability Statement:** The data presented in this study are available in the main article.

**Acknowledgments:** The authors gratefully acknowledge the financial support of the National Key Research and Development Project of China (2017YFE0125400) and the National Natural Science Foundation of China (51901048).

**Conflicts of Interest:** The authors declare no conflict of interest.

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
