# Peer review of "A Comparative Investigation on the Microstructure and Thermal Resistance of W-Film Sensor Using dc Magnetron Sputtering and High-Power Pulsed Magnetron Sputtering"

_magnetochemistry, doi:10.3390/magnetochemistry9040097_

Round 1
Reviewer 1 Report
Several issues should be clarified/corrected before publication: 1. Uncommon abbreviations (such as dcMS and HiPIMS) should be avoided in title. 2. Lines 149-151. Please, indicate a base pressure and Ar flow rate in the case of dcMS sputtering. 3. Lines 156-157. English should be corrected. 4. Resistivity measurements should be described in details: 2- or 4- wire strategy was used? Distance between electrodes? How electrodes were connected to the W layer. 5. Fig. 2. Difference in morphology between dcMS (grains) and HiPIMS (almost flat) should be commented. 6. Lines 199-200, the phrase "in terms of the overall shape of the surface section" should be clarified. 7. Fig. 3 Z-scale is not clearly seen. 8. Lines 232-245. How zero shift was corrected? Difference in peak height may be corresponded with different thickness of the W films. "the XRD half-height length of the metal tungsten film prepared by the HiPIMS process is larger, which also shows that the film prepared by the HiPIMS process is more refined" - what this mean? 9. Fig. 5. Colors should be unified for left and right columns. 10. Lines 326-331. English should be corrected. 11. Fig. 11. Right shift is not clearly seen. Authors are advised to plot the relative peak position vs annealing conditions. 12. Scale of resistivity and colors should be unified with Fig. 5.Author Response
Please see the attached document.

Reviewer 2 Report
The authors report on growth and characterization of tungsten films grown either by dc or high power pulsed magnetron sputtering. This is a rather comprehensive study and might be of interest to researchers in the field.
Overall, the paper is well written. However, I find the discussion of the XRD data a bit superficial. First, it would be beneficial to show the data in Figs. 4, 8 and 11 on a semilog scale. The authors conclude from a left-shift of the (110) peak on the presence of compressive stress. It would be useful to quantify the strain. Further, what can be said about the half-width of the peaks? There appears to be a dependence on deposition method and pulse length, but it is difficult to discern from the figures. This should be discussed in more detail.
I recommend publication after minor revisions.
Reviewer 3 Report
The manuscript is interesting, it compares the Tungsten thin film deposited by two methods and then applied as a temperature sensor. Morphological, structural, and related parameters to the desired application are presented. The manuscript is clear, well organized, and designed, the results are supporting each other and the story is well written. No comments from my side.
Round 2
Reviewer 1 Report
All reviewer's suggestions have been revised and/or commented successfully.
The manuscript can be accepted in present form.